# Emergence of Integrated Information at Macro Timescales in Real Neural Recordings

**DOI:** 10.3390/e24050625

**Published:** 2022-04-29

**Authors:** Angus Leung, Naotsugu Tsuchiya

**Affiliations:** 1Turner Institute for Brain and Mental Health, School of Psychological Sciences, Monash University, Melbourne, VIC 3800, Australia; 2Center for Information and Neural Networks (CiNet), National Institute of Information and Communications Technology (NICT), Suita 565-0871, Japan; 3Advanced Telecommunications Research Computational Neuroscience Laboratories, Soraku-gun, Kyoto 619-0288, Japan

**Keywords:** integrated information, anaesthesia, emergence, *Drosophila*, consciousness, information, integration, time, temporal

## Abstract

How a system generates conscious experience remains an elusive question. One approach towards answering this is to consider the information available in the system from the perspective of the system itself. Integrated information theory (IIT) proposes a measure to capture this integrated information (Φ). While Φ can be computed at any spatiotemporal scale, IIT posits that it be applied at the scale at which the measure is maximised. Importantly, Φ in conscious systems should emerge to be maximal not at the smallest spatiotemporal scale, but at some macro scale where system elements or timesteps are grouped into larger elements or timesteps. Emergence in this sense has been demonstrated in simple example systems composed of logic gates, but it remains unclear whether it occurs in real neural recordings which are generally continuous and noisy. Here we first utilise a computational model to confirm that Φ becomes maximal at the temporal scales underlying its generative mechanisms. Second, we search for emergence in local field potentials from the fly brain recorded during wakefulness and anaesthesia, finding that normalised Φ (wake/anaesthesia), but not raw Φ values, peaks at 5 ms. Lastly, we extend our model to investigate why raw Φ values themselves did not peak. This work extends the application of Φ to simple artificial systems consisting of logic gates towards searching for emergence of a macro spatiotemporal scale in real neural systems.

## 1. Introduction

Integrated information theory tackles the question of how physical interactions can support consciousness by introspecting conscious experience [1,2]. It then deduces postulates, the necessary physical interactions to support conscious experience, and from these derives a numerical measure of consciousness which should be high in a conscious system, and low otherwise. We previously applied the measures proposed by integrated information theory (IIT) 3.0 to local field potentials (LFPs) from the fly brain, testing the hypotheses that system-level integrated information Φ and its associated conceptual structure should be reduced during reduced level of consciousness as induced by anaesthesia [3]. As expected from the theory, both Φ and associated conceptual structures computed from the LFPs were indeed reduced during anaesthesia. However, we were unable to apply all of IIT’s postulates exactly as they are put forward by the theory. Specifically, we did not fully apply IIT’s exclusion postulate, which states that only one set of overlapping sets of elements, the complex, can be conscious.

To identify the complex, IIT’s exclusion postulate requires searching across all subsets of system elements, recomputing Φ for each subset. However, this search quickly becomes computationally infeasible for larger numbers of elements, due to the rapidly increasing cost of repeatedly identifying the minimum information partition (MIP; [2]) for all subsets. IIT’s exclusion postulate also requires searching for the complex across spatial and temporal scales. As LFPs are an aggregate measure which summate electrical activity arising from neurons’ cell bodies, axons and dendrites at a scale much coarser than that of individual neurons [4], searching for the potential spatial scale of the complex did not seem to be a promising avenue for investigation. However, searching for the temporal scale is a feasible and likely more fruitful endeavour, given the high temporal resolution of LFPs.

IIT provides a clear expectation as to the temporal scale of the conscious complex. Specifically, IIT’s exclusion postulate ties the complex to the scale at which our experiences occur. Through introspection, it is apparent that, for humans, an instance of experience occurs most likely at the scale of milliseconds—we are unable to perceive events which occur at too short a timescale, such as events which occur at the scale of microseconds. This intuitive scale is backed empirically by psychophysics studies, with humans being able to discern events at the scale of tens of milliseconds, but not shorter [5]. It is also unlikely to be at longer timescales such as seconds or longer, where we can differentiate multiple instances of experience. Consequently, through its exclusion postulate, IIT predicts that Φ should be maximal at some particular scale, in the order of milliseconds. Conversely, it should be lower, both at micro timescales which are too short and at macro timescales which are too long to correspond to the timescale of conscious experience. While the emergence of maximal Φ at some intermediate scale has been previously illustrated in example binary systems consisting of logic gate elements [6,7,8], it is unclear whether it occurs in real neural data which is typically continuous in nature.

In this paper, we will test the above prediction using both real neural recordings and related computational generative models. First, by using a toy auto-regressive model, for which temporal interactions among system elements are known a priori [9], we verify that Φ identifies the timescale of system interactions from continuous data generated by the model. Next, we apply Φ to the fly recordings previously analysed in [3], to search for a potential temporal scale of interactions in the complex. However, we find Φ to either increase or decrease in a monotonic fashion with changes in temporal scale both when flies were wakeful and anaesthetised, depending on how the recordings were pre-processed to characterise timescale. Meanwhile, the ratio of wakeful to anaesthetised Φ identifies a potential temporal scale of interactions, again depending on how timescale is characterised. Given these findings, in the last section of this paper, we expand the auto-regressive simulation to explore limitations of our application of Φ to the fly recordings, namely non-Markovianity and partial observation.

## 2. Results

### 2.1. Integrated Information Identifies the Timescale of Interactions in a Nonlinear Autoregressive Process

Example interactions between system elements leading to maximal Φ occurring not at the most fine-grained scale, but rather at a macro scale, have previously been illustrated in toy systems with binary elements. However, this illustration has not been extended to systems with continuous elements. So, to check the in-principle feasibility of searching for emergence of Φ at a macro scale in continuous data, we first utilised a toy autoregressive model, where the value of each time-sample is determined by values at previous times [10]. We modelled a bivariate, bidirectionally connected system as follows:$$X\left(t\right)=aX\left(t-l\right)+bY\left(t-l\right)+\varepsilon_{X}\left(t\right)$$
$$Y\left(t\right)=cY\left(t-l\right)+dX\left(t-l\right)+\varepsilon_{Y}\left(t\right)$$
where *X*(*t*) and *Y*(*t*) are voltages for two system elements (which we refer to as channels) at a given time *t*. *a* and *c* are autoregressive coefficients representing self-connections, while *b* and *d* are autoregressive coefficients representing cross-connections between the two elements *X* and *Y*. We set both *a* and *c* as −0.1, simulating self-inhibition. *l* is the time delay between self- and cross-connections among system elements, which we set to be 10. ε*_X_*(*t*) and ε*_Y_*(*t*) represent uncorrelated Gaussian noise, with mean 0 and variance both set to 0.5.

As the neural mechanisms underlying LFPs are known to have nonlinear dynamics, we included nonlinearity in the model. We set the cross-connectivity to be dependent on the voltages of *X* and *Y*:$$b=\{\begin{array}{cc} 0.9, & Y\left(t-l\right)>threshold\\ 0, & otherwise \end{array}$$
$$d=\{\begin{array}{cc} 0.9, & X\left(t-l\right)>threshold\\ 0, & otherwise \end{array}$$
where *threshold* was set as 0.9. This adds a nonlinear dynamic which simulates reliable neural communication through bursting [11,12]. Note that, overall, the system elements are bidirectionally connected, and only interact with a delay of 10 timesteps. Consequently, we expected Φ to be non-zero for this system (which we previously reported for *l* = 1 in [3], S7 Text), and critically, maximal at the timescale corresponding to 10 timesteps.

To check that Φ does indeed identify this timescale of 10 timesteps, we simulated the model for 10 runs (see Section 5). For each run we operationalised the state of each channel at a given time point by binarising it with respect to the median voltage of that channel. Then we constructed a transition probability matrix (TPM) by finding, for each state of the system at time *t*, the empirical probabilities of each channel being in an “on” state at *t* + τ. This is the same method we previously used to compute TPMs for real neural recordings in [3] (see also Discussion in [3] for issues regarding observation versus perturbation in constructing TPMs). From this method, which we hereafter refer to as the “skipping” method, timescale is characterised as the delay τ. We repeatedly computed TPMs for exponentially increasing values of τ (Figure 1B,C) and, from these TPMs, computed Φ values at each τ value.

Figure 1F shows the trend of Φ with respect to τ when using the skipping method. While there existed multiple local maxima of Φ (peaks at roughly τ = 10, 60 and 360 timesteps), Φ was, as expected, maximal at τ = 10 timesteps, corresponding to the time delay *l* = 10 in the model.

While the skipping method is consistent with how empirical estimates of Φ from previous versions of IIT have been applied [13,14,15], simulation papers illustrating maximal Φ at macro temporal scales have utilised different methods [6,7,8]. Specifically, they utilise coarse-graining or black-boxing, whereby micro timesteps are collated together to form macro timesteps. Following this approach, we characterised timescale in a second way, by averaging voltages in bins of size τ (green and blue rectangles in Figure 1B,D). Then, in the same manner as to the skipping method, we operationalised states of each channel by binarising the resulting downsampled voltages based on their medians to construct TPMs for increasing τ (Figure 1E). We refer to this method as the “downsampling” method.

The downsampling method has a notable drawback compared to the skipping method. Specifically, for a given time-series, when τ is increased by some factor, the number of time-samples available for building a TPM is decreased by that factor. For example, doubling τ would result in half the original number of samples in the time-series. Due to the fewer number of samples and thus fewer overall state transitions, empirical transition probabilities in the TPM can rapidly become unreliable as τ increases. To address this, we constructed TPMs from multiple rounds of downsampling, by offsetting the starting time sample of each bin (Figure 1D) before downsampling and then binarising voltages. In this manner, a TPM for a given τ was constructed using all transitions from all offsets. Using this method, the number of transitions used to construct a TPM was equal to the number of transitions used in the skipping method.

Figure 1G shows the trend of Φ with respect to τ when using the downsampling method. While Φ seemed to be non-minimal for a larger range of τ when compared to the skipping method, it was, again as expected, maximal at τ = 10, corresponding to the time delay *l* = 10 in the model. These results indicate that, Φ identifies the timescale of interactions among continuous processes both when using the skipping and downsampling methods.

### 2.2. Normalised Empirical Integrated Information Identifies a Timescale of Interactions

We next sought to find some timescale in neural recordings at which Φ is maximised. We utilised 15 local field potentials (LFPs, hereafter referred to also as “channels”) recorded from across the brains of 13 fruit flies using a linear multi-electrode array as previously described in [3,16].

As we did previously for the simulation, we operationalised the state of each channel by binarising voltages based on the median voltage for the channel before then constructing a TPM at increasing values of τ (skipping method), as well as by repeatedly binarising voltages based on median voltages after downsampling at increasing values of τ (downsampling method). Given the computational cost of computing Φ, and needing to repeatedly compute Φ at each τ value, we restricted analysis to two channels at a time, treating every pair of channels as a system.

Figure 2 shows the trend of Φ (log transformed) with increasing τ across the flies, for both methods of characterising timescale. On average across all the channel pairs, there was no visual indication of Φ being maximal at a timescale other than the smallest or largest timescales. Rather, Φ trended such that it tended to be larger for smaller timescales when using the skipping method (Figure 2A), and larger for larger timescales when using the downsampling method (Figure 2B).

As a control, we also computed Φ for the channel pairs when the flies were anaesthetised. We reasoned that, during loss of consciousness, Φ should not have a clear maximum at some timescale. Rather, assuming that there is no consciousness under anaesthesia, it should be minimal at all timescales. Any variations in Φ across timescale should correspond not to a potential complex of consciousness, but instead to other things such as background neural activity which does not support consciousness (or supports some minimal consciousness) or issues regarding empirical observations of state transitions which are used to build the TPM (which we expand on in the Discussion). Blue lines in Figure 2A,B show the trend of Φ with increasing τ during anaesthesia respectively when using the skipping and downsampling methods. While the magnitude of Φ tended to be overall reduced across all τ when compared to wakefulness, consistent with our previous results [3], the trends of Φ with respect to τ, for both the skipping and downsampling methods, appeared to be the same as for wakefulness.

Given that the trends of Φ with respect to τ during anaesthesia was similar to during wakefulness, we considered that the trends during wakefulness could also be reflecting issues of empirical observation of TPMs. Meanwhile, any trend of Φ related to the timescale of interactions underlying the complex could be masked by these trends. To address this, we considered using Φ values during anaesthesia as a baseline. Specifically, we investigated how the difference (wake minus anaesthesia) in log transformed Φ values (Δlog(Φ); corresponding to taking the ratio of wakeful to anaesthetised values in the natural scale) varied with τ.

Figure 2C,E shows the trend of Δlog(Φ) across τ when using the skipping method. Unlike raw Φ values, visual inspection indicated a peak of Δlog(Φ) in the range of τ = 8 to τ = 16 ms. While this peak was most prominent for the most centrally located channel pairs, it appeared to extend across the fly brain. To confirm that there was indeed a peak within this range of τ, we utilised mixed effects analysis (to account for intra-fly channel pair correlations, see Section 5), regressing Δlog(Φ) onto a quadratic term τ^2^. The turning point of the fitted quadratic would indicate a peak of Δlog(Φ) at some timescale other than the smallest or largest ones if: (1) it is a local maximum (corresponding to the fitted coefficient for τ^2^, β_2_, being negative) and (2) it occurs at some intermediate timescale. We first statistically confirmed previous visual inspection that no such peak occurred in the raw Φ values during wakefulness or anaesthesia (Table A1). Meanwhile, the observed peak in Δlog(Φ) when using the skipping method was indeed statistically significant, with fitted coefficients indicating a local maximum at roughly 5 ms (χ^2^(1) = 663.99, β_2_ = −9.18 × 10^−3^, β_1_ = 4.441 × 10^−2^, β_0_ = 0.433).

We next checked if this result could also be found using the downsampling method (Figure 2D,F). Given the previous simulation results, we expected to find a similar peak to when using the skipping method. However, visual inspection indicated that the greatest Δlog(Φ) occurred at the smallest τ. The lack of a peak at some intermediate timescale was statistically confirmed by a positive regression coefficient for regressing Δlog(Φ) onto τ^2^ (Table A1). 

### 2.3. Integrated Information Identifies the Timescale of Interactions under Non-Markovianity

Though we found some indication of a temporal peak, for Δlog(Φ) when using the skipping method, we were unable to identify such a peak in the raw Φ values themselves, or for Δlog(Φ) when using the downsampling method. So, we next considered whether particular limitations regarding the application of IIT to neural data could have directly prevented any such finding. Specifically, we considered the limitations which we previously identified in [3] regarding the validity of Φ when there is potential of spurious correlations among system elements, which can occur in non-Markovian systems and when multivariate systems are only partially observed.

We first investigated the issue of non-Markovianity. Specifically, non-Markovianity may be problematic for Φ as IIT 3.0 is entirely constructed for Markovian systems where the state of a system depends only on its immediate previous state. To test if non-Markovianity immediately invalidates the application of Φ with regards to identifying the timescale of system interactions, we extended the previous nonlinear autoregressive model by modifying the lag term *l*. Specifically, we set the lag term *l* to be jittered among 9, 10 and 11 in a probabilistic manner. This way, the system cannot be described as a purely Markovian system where its state at time *t* is completely determined by its state at time *t* − *l* for some fixed *l*. For simulation, we initialised processes *X* and *Y* to uncorrelated Gaussian noise with mean 0 and variance both set to 0.5:$$X\left(t\right)=\varepsilon_{X}\left(t\right)$$
$$Y\left(t\right)=\varepsilon_{Y}\left(t\right)$$

Then, for each timepoint *t*:$$aX\left(t\right)\to X\left(t+l_{a}\right),\;bY\left(t\right)\to X\left(t+l_{b}\right)$$
$$cY\left(t\right)\to Y\left(t+l_{c}\right),\;dX\left(t\right)\to Y\left(t+l_{d}\right)$$
where “→” denotes updating the right-hand value by adding the value on the left. We added non-Markovianity here by probabilistically choosing *l_a_*, *l_b_*, *l_c_* and *l_d_* to be 9, 10, or 11 timesteps, all independently of one another, with probability 0.25, 0.5 and 0.25 respectively. Consequently, each time sample could have been determined by either 1, 2, or 3 individual timepoints from the past. This simulates variability in neural spike or burst timings [19,20]. Note that, while the model is now non-Markovian, the system elements still clearly interact at a timescale of roughly 10 timesteps. The cross-connection strengths were, as for the first simulation, dependent on a threshold voltage:$$b=\{\begin{array}{cc} 0.9, & Y\left(t\right)>threshold\\ 0, & otherwise \end{array}$$
$$d=\{\begin{array}{cc} 0.9, & X\left(t\right)>threshold\\ 0, & otherwise \end{array}$$
with *threshold* again being 0.9.

Figure 3 shows the trend of Φ when computed from the time-series generated by this model, for both the skipping and downsampling methods. For both methods, Φ was maximal at the scale of 10 timesteps, corresponding to the timescale of system interactions. Hence, non-Markovianity per se does not prevent Φ from identifying the timescale of system interactions. However, non-Markovianity did appear to affect the magnitude of Φ values when using the skipping method. Specifically, Φ was an order of magnitude lower than in the first simulation (maximum Φ being ~0.025 in Figure 3A, compared to ~0.13 in Figure 1F). This drastic reduction in Φ did not occur when using the downsampling method (Figure 3B).

### 2.4. Integrated Information Identifies the Timescale of Interactions under Partial Observation

Given that we were able to identify the timescale of interactions even with a violation of Markovianity, we next turned towards the issue of partial observations. As Φ computed from partial observations (i.e., not across the full system, or complex) is not postulated to correspond to consciousness per se, it could be the case that the timescale is smoothed out through delayed effects from interactions from non-observed system elements. To test this, we extended our non-Markovian system by introducing a third element, giving a total of three system elements. Specifically:$$X\left(t\right)=\varepsilon_{X}\left(t\right)$$
$$Y\left(t\right)=\varepsilon_{Y}\left(t\right)$$
$$Z\left(t\right)=\varepsilon_{Z}\left(t\right)$$
*Z*(*t*) is the third system element, and ε*_Z_*(*t*) represents Gaussian noise, with mean 0 and variance both set to 0.5. Then, at each timepoint *t*, time samples affected future time points through self- and cross-connections with some lag *l*:

$$aX\left(t\right)\to X\left(t+l_{a}\right),\;bY\left(t\right)\to X\left(t+l_{b}\right),\;gZ\left(t\right)\to X\left(t+l_{g}\right)$$$$cY\left(t\right)\to Y\left(t+l_{c}\right),\;dX\left(t\right)\to Y\left(t+l_{d}\right),\;hZ\left(t\right)\to Y\left(t+l_{h}\right)$$$$eZ\left(t\right)\to Z\left(t+l_{e}\right),\;fX\left(t\right)\to Z\left(t+l_{f}\right),\;iY\left(t\right)\to Z\left(t+l_{i}\right)$$
where *e* is the self-connection of *Z*, and *f*, *g*, *h* and *i* are the new cross-connections connecting all three system elements *X*, *Y,* and *Z* bidirectionally. In this model, we set all self-connections *a*, *c*, and *e* to −0.1, and all cross-connections to 0.4. All the lag terms *l*_a–i_ were independent and probabilistic, taking values again of 9, 10, or 11 with probabilities 0.25, 0.5 and 0.25. Again, cross-connection strengths were dependent on a threshold voltage

$$\pi =\{\begin{array}{cc} 0.4, & \Pi \left(t\right)>threshold\\ 0, & otherwise \end{array}$$
where π is a cross-connection coefficient (*b*, *d*, *f*, *g*, *h*, or *i*), and Π(*t*) is the voltage for the associated channel (*X*(*t*), *Y*(*t*), or *Z*(*t*); e.g., for cross-connection π = *b*, the associated channel is Π(*t*) = *Y*(*t*)). *threshold* was again 0.9, for all connections.

We first confirmed that our previous findings regarding Φ identifying the timescale of system interactions in the two channel case extends to the three channel case (Figure 4A,B), by computing Φ for two channels at a time. Ideally, background conditions (i.e., the states of channels outside those being used to compute Φ) should be fixed. However, in real neural data, doing so drastically limits the number of observations available to build a TPM. Further, the number of possible background conditions to consider grows exponentially with the number of channels. Consequently, fixing background conditions to compute Φ is infeasible for real neural data, and so we also did not fix background conditions in this simulation. As expected, Φ was maximal at the timescale corresponding to 10 timesteps. Though the magnitude of Φ at this peak was lower than in the previous 2-channel simulations, this was expected from a fully connected system. Specifically, system states in a fully connected system have low specificity about their causes and effects, and this should result in low Φ [21,22]. Though our 2-channel simulations were also fully connected, the only other way of connecting two channels is using a unidirectional connection, which would result in minimal Φ (see [3] S7 Text).

To test whether partial observation prevents Φ from identifying the timescale of system interactions, we then computed Φ on two channels, out of the three, at a time. This simulates the case of not being able to observe the states of all neurons in the brain. Or, as previously in the fly LFPs, the case of not being able to compute Φ using all available observations. Figure 4A,B also show the trend of Φ when computed from two channels at a time, in relation to timescale. Similar to non-Markovianity, the magnitude of Φ was again reduced by an order of magnitude, this time for both the skipping and downsampling methods. However, Φ was still maximal at the timescale of 10 timesteps, suggesting that partial observation per se also does not in principle prevent Φ from identifying the timescale of system interactions.

## 3. Discussion

Here we applied the measure Φ to simple autoregressive models and real neural data, both with continuous system elements. Φ has been proposed by integrated information theory 3.0 (IIT) to be maximal at a temporal scale corresponding to that of conscious experience. Here, we demonstrated that for a nonlinear system, Φ can be maximal to the timescale corresponding to that at which system elements interact. We also applied Φ to neural data, finding that the measure, when normalised, peaks at a timescale of roughly 5 ms. Finally, in follow-up simulations we demonstrated that Φ still peaks at the timescale at which system elements interact, even when certain assumptions of IIT, namely Markovianity and full observation of the system, are not met.

The emergence of a temporal peak of Φ has previously been illustrated in simulation studies utilising systems consisting of binary elements [6,23]. These studies focused on utilising the framework provided by IIT to question the common view posed by reductionism—that the causal structure of a system is fully captured at the most fine-grained level, with there being no room for causal contribution from macro spatiotemporal scales. Rather, they posit that Φ can capture and describe causal emergence, whereby interactions at a macro scale contribute to the causal structure of a system beyond those at the most fine-grained level. The simulation results presented here extend their illustration of causal emergence across temporal scales, as captured by Φ, to systems with continuous elements.

### 3.1. Why Is There a Peak in Normalised Φ but Not Directly in Φ?

Though we found Φ to clearly peak at the timescale of interactions among system elements in the autoregressive models, we observed no such peak in fly LFPs during wakefulness or anaesthesia. Instead, we found a temporal peak to manifest for normalised Φ, the ratio of Φ during wakefulness to anaesthesia. Why this was the case is not yet clear, but there are some considerations which may have prevented Φ from clearly peaking at some intermediate timescale, as was the case for Δlog(Φ) when using the skipping method to characterise timescale.

One potential explanation regards the effects of non-Markovianity and partial observation. While the peaks in Φ for the simulated systems reliably matched the timescale at which their elements interacted with one another, the systems were designed to have clear temporal dynamics. Specifically, elements interacted with a consistent delay of around 10 timesteps. However, the temporal dynamics of the brain are much less clear, where the effects of non-Markovianity and partial observation are likely to be much greater than in the models used here. For example, autoregressive models fit to LFPs from monkeys have been fit to the 10th or 20th order with timesteps of 5 ms [24,25], with many historical timesteps potentially influencing any one given time sample. Consequently, any one ideal temporal scale may be greatly blurred. Indeed, in the simulations here, peak Φ values reduced as non-Markovianity and partial observation were incrementally added to the models. As these factors are present both during wakefulness and anaesthesia, it is conceivable that normalising wakeful Φ by anaesthetised Φ cancels them out to some extent.

A second potential explanation regards the TPMs used for computing Φ, which were constructed at each timescale. For a given TPM, the number of transitions used to construct it depended on its associated timescale. Specifically, for *n* time samples, the number of transitions that can be used to construct the TPM is *n* − τ when using the skipping method, and *n* − τ − 1 when using the downsampling method. Consequently, each entry of the TPM is determined using fewer samples as τ increases, with probabilities becoming less reliable and more likely to take more deterministic values (i.e., probabilities closer to 0 or 1). This in turn may cause Φ values to increase systematically with τ, as more deterministic probabilities allow for greater information in each system state. While we observed this trend for the downsampling method, the skipping method however revealed an opposite trend. At this point, it is unclear how less reliable but more deterministic seeming TPMs would result in both increasing and decreasing Φ values, depending on the method used to characterise timescale. However, the systematic effect may further hide the temporal scale of a system, while meanwhile being cancelled out by normalising wakeful Φ values by anaesthetised Φ values.

### 3.2. Why Do Skipping and Downsampling Methods Give Different Peaks?

The autoregressive simulation results presented here indicated that Φ would be maximal at the timescale corresponding to that at which system elements interact, regardless of whether the skipping or downsampling methods were used. Specifically, Φ computed from both methods should identify the same timescale. However, this was not the case in fly data for Δlog(Φ), where Δlog(Φ) peaked at roughly 5 ms when using the skipping method but not the downsampling method. While it is not immediately clear as to why only one method would identify a peak, here we provide a potential interpretation of this result.

While the simulations we used here had very clear dynamics at a particular, specific timescale, it is conceivable that interactions in the brain take place at multiple timescales. Multiple timescales may exist by virtue of the skipping and downsampling methods capturing different types of timescales. Specifically, the skipping method captures the delay between system elements being in some particular state affecting others. An example of different timescales of this type might be short and long range connections having shorter and longer delays respectively. Meanwhile, the downsampling method instead tries to capture the temporal size of the states the system elements can take. Different timescales of this type could manifest as, for example, both neuronal bursting and individual neuronal spikes being states which influence other neurons. Further simulations incorporating the above considerations may be required to understand how Φ or Δlog(Φ) behaves when system elements interact across multiple such timescales.

Taking into account the above considerations, the peak in Δlog(Φ) computed using the skipping method at 5 ms may reflect just one timescale at which neuronal interactions occur. This timescale sits between two neurophysiologically reported timescales. The first is that of axon conduction delays, the delay in firing between connected neurons, which is known to be on the order of single-digit milliseconds [26]. The second is that of critical flicker fusion frequency, the frequency at which a flickering visual stimulus is indistinguishable from a constant stimulus. For flies, the critical flicker fusion frequency has been reported, using electroretinograms, to be at 57 Hz [27], with each individual flicker lasting ~18 ms. We note however that critical flicker fusion frequencies are known to vary, at least in humans, depending on a variety of factors such as stimulus size and intensity and perceptual load [28,29], and that flicker fusion frequencies have not to our knowledge been validated in flies using a behavioural paradigm.

Meanwhile, Δlog(Φ) computed using the downsampling method peaking in the shorter timescales (1–2 ms; Figure 3) may correspond more directly to the shorter timescale of axon conduction delays. While regressing Δlog(Φ) computed using this method onto timescale did not reveal a negative parabolic trend with a global maximum, this may have been due to not having higher sampling rate data. Thus, it is unclear whether this peak is a potential global maximum or just a general trend of Δlog(Φ) increasing with shorter timescales.

## 4. Conclusions and Future Directions

This work is to our knowledge the first direct application of IIT to search for a potential timescale of consciousness using neural data. While a previous study characterised a proxy measure of Φ, Φ_AR_, across timescales in electroencephalographic recordings from infants [30], the Φ values were negative for most of the timescales investigated, making their interpretation unclear within the framework of the theory [14]. Meanwhile, here we identified a timescale which aligns with neural physiology and potentially flies’ behaviourally and phenomenologically (if any) relevant flicker fusions. However, this comes with the caveat that raw Φ values from the fly recordings either increased or decreased monotonically across timescales, depending on the pre-processing method used. Consequently, more work, utilising both simulation and neural recordings with higher temporal and spatial resolution, is required to confirm whether Φ peaks uniquely at this identified timescale or at varying timescales depending on the method used for characterising timescale. Within this line of work, other methods of characterising timescale should be explored in neural data, such as grouping micro states with logical operations or through black-boxing [6,7]. There is also the further question of whether the peak identified here persists across differing spatial scales, such as at the single neuron level. Finally, behavioural paradigms which capture the temporal scale of conscious experience in a system would be required to more strongly link this potential peak in Φ to consciousness.

## 5. Methods

As the fly LFPs analysed here are the same data as described and analysed in [3], we refer the reader there instead of repeating the details here. Details regarding the algorithm for computing Φ from TPMs are also identical to those provided in [3] (albeit for two channels at a time, instead of four channels, due to the extra computational cost of repeatedly computing Φ at different timescales). So, here we provide only the details regarding generating data from the autoregressive models described in the Results section, and statistical analyses of the LFPs.

### 5.1. Autoregressive Simulation

Model simulation and data analyses were conducted using MATLAB 2019b. For each of the three autoregressive models (each additionally including nonlinearity, non-Markovianity and partial observations), we simulated 20,000 timepoints, for each of 10 runs. The initial conditions for each run were determined by the uncorrelated noise terms ε*_X_*, ε*_Y_* and ε*_Y_*, as described in the Results.

### 5.2. Φ Computation

Data processing for computing Φ was conducted using Python 3.6.0 in MASSIVE (Multi-modal Australian ScienceS Imaging and Visualisation Environment), a high-performance computing facility. We calculated the measures using PyPhi (version 0.8.1; [31]), publicly available at https://github.com/wmayner/pyphi (accessed on 7 March 2022). Detailed description regarding the computation of Φ from the TPM, are provided in [2,3,31].

### 5.3. Statistical Analyses

We used linear mixed effects analysis (LME; [32,33]) to statistically test for a peak in Φ at some intermediate timescale (i.e., not corresponding to the shortest or longest timescales). This allowed us to account for within-fly correlations among channel pairs without averaging across channel pairs or flies, by including random intercepts for fly and the interaction between fly and pairs as random effects. As Φ was positively skewed, we analysed log transformed values. To test for a potential peak in Φ at some intermediate timescale, we first assessed the significance of a quadratic fit by comparing the quadratic models (Table A2) to models with only a linear term (in Wilkinson notation [33]):Φ ~ τ + (1|*fly*) + (1|*fly*:*pair*)
where Φ is one of Φ*_SW_*, Φ*_SA_*, Φ*_S_*_Δ_, Φ*_DW_*, Φ*_DA_*, Φ*_D_*_Δ_, and τ is timescale (see Table A2). Subscripts *S* and *D* indicate Φ computed using the skipping and downsampling methods, respectively, while *W* and *A* indicate Φ computed during wakefulness or anaesthesia. Subscript Δ indicates Δlog(Φ). As we searched through exponentially increasing τ, we fitted to log_2_(τ) values. To assess significance, we used likelihood ratio tests, comparing the log-likelihood of the quadratic model to the log-likelihood of the linear model. As the likelihood ratio is χ^2^ distributed when one model is nested within another with degrees of freedom equal to the difference in number of coefficients between the models, we report χ^2^(*d*.*o*.*f*) and its corresponding *p*-value. We summarise the amount of variance explained by each fitted quadratic model, and by each random effect, in Table A2.

Given that the quadratic term significantly increased the variance explained by the model, we then calculated the turning points of each fitted model. We considered there to be an intermediate peak in Φ if the turning point was a local maximum and occurred at some τ other than the most extreme timescales.

## Figures and Tables

**Figure 1 entropy-24-00625-f001:**
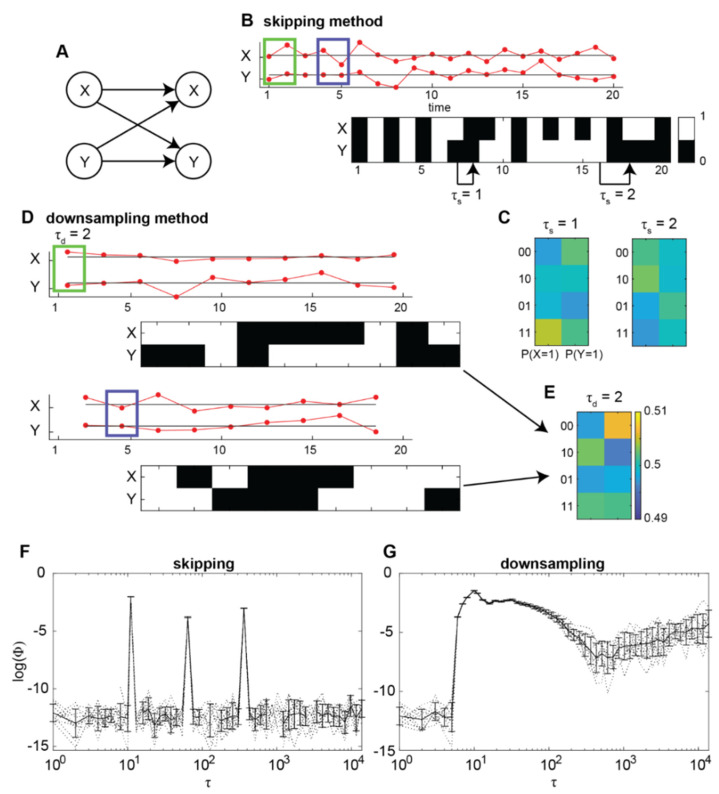
Relationship between integrated information Φ and timescale τ in a system with nonlinearity. (**A**) We generate continuous time-series by modelling a nonlinear, bidirectionally connected system. (**B**) For the skipping method, continuous time-series values (red, top) are discretised into binary states (black/white, bottom) by comparing to the median value for each run. Displayed is an example of 20 samples from 1 run. (**C**) State-by-node transition probability matrices (TPM) are constructed using the skipping method for increasing τ. For each possible system state at time *t* (each row in the TPM), each entry describes the probability a node will take state “1” at time *t* + τ. (**D**) For the downsampling method, τ contiguous time-series values are averaged together to form coarse-grained time-series. Multiple downsampled time-series are obtained by offsetting the time sample from which to begin coarse-graining, from 0 up to τ − 1 samples. Green rectangles indicate the first bin of contiguous time samples, for τ = 2, from the original time-series in (**A**) which are averaged together, for the first offset (of 0 samples). Blue rectangles indicate the second bin for the second offset. Coarse-grained time samples are then discretised into binary states by comparing the median value for each offset, at each run. (**E**) TPMs are constructed for the downsampling method using all transitions across all offsets. Each entry describes the probability a node will have a coarse-grained state “1” at a coarse-grained time *t* + 1, given the system state at *t*. (**F**) Φ values in relation to τ when using the skipping method. Dotted and solid lines indicate individual simulation runs and the average across runs, respectively. Error bars indicate standard deviation across runs. (**G**) Same as (**F**), but for Φ values computed using the downsampling method.

**Figure 2 entropy-24-00625-f002:**
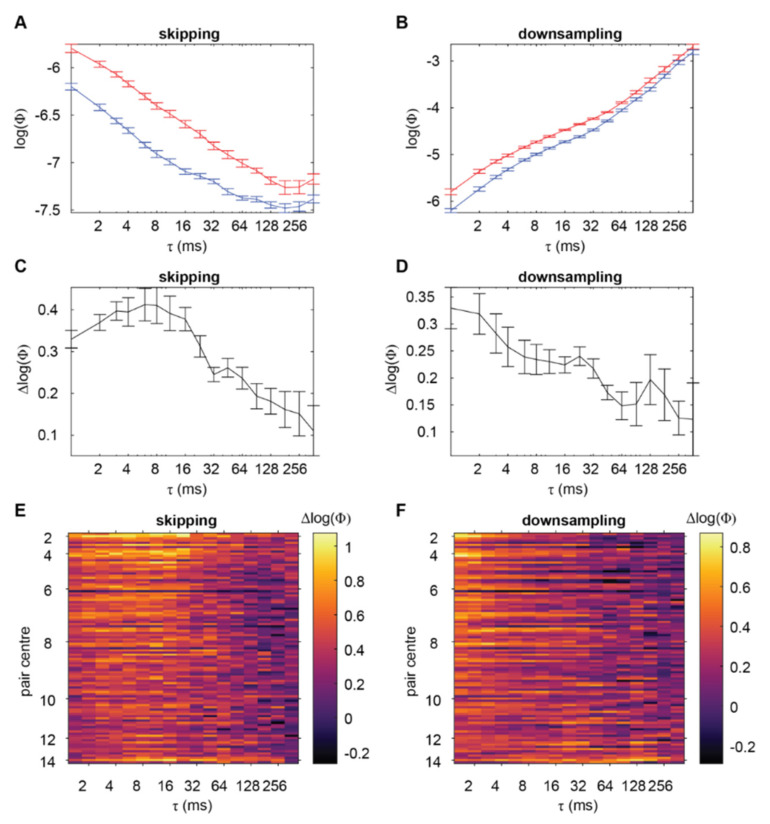
Relationship between integrated information Φ and timescale τ in fly recordings. (**A**) Log transformed Φ values, averaged across channel pairs and flies, as a function of timescale when using the skipping method. Red and blue are values during wakefulness and anaesthesia, respectively. Error bars indicate within-subject standard error [17,18]. (**B**) Log transformed Φ values, as in (**A**), but for when using the downsampling method. (**C**,**D**) Difference between wakeful and anaesthetised log transformed Φ, Δlog(Φ), as a function of timescale when using the skipping and downsampling methods respectively. (**E**,**F**) Δlog(Φ) as a function of timescale for each channel pairing when using the skipping and downsampling methods respectively. Channel pairs are sorted by the average position of the channels in the pair (*y*-axis, larger values indicate pairs which on average are located more in the periphery). Pairs with the same average position are sorted by the distance between the channels, with larger distances being lower in the *y*-axis. τ (*x*-axis) increases in an exponential manner.

**Figure 3 entropy-24-00625-f003:**
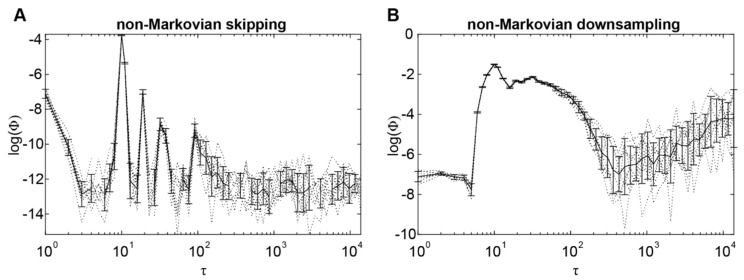
Relationship between integrated information Φ and timescale τ in a nonlinear system (Figure 1) extended with non-Markovianity. (**A**) Log transformed Φ values in relation to τ when using the skipping method. Dotted and solid lines indicate individual simulation runs and the average across runs, respectively. Error bars indicate standard deviation across runs. (**B**) Same as (**A**), but for Φ values computed using the downsampling method.

**Figure 4 entropy-24-00625-f004:**
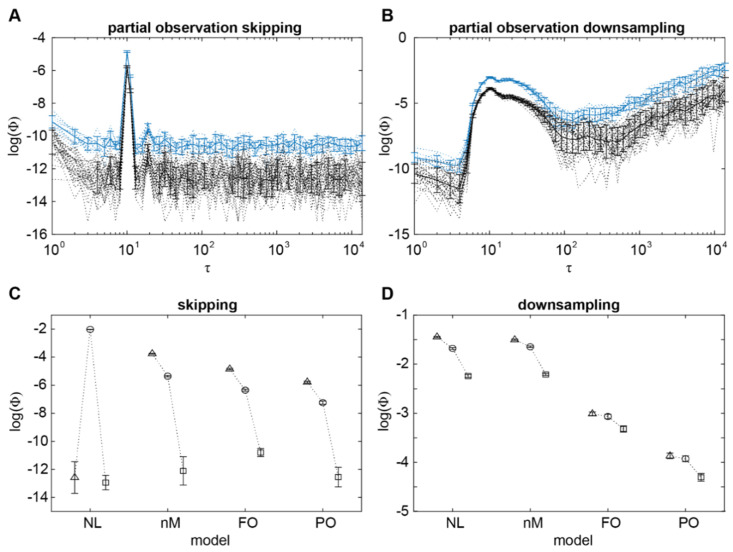
Relationship between integrated information Φ and timescale τ in a system with nonlinearity and non-Markovianity, under partial observation. (**A**) Log transformed Φ values computed from all three channels (full observation; blue), and for values computed from two channels at a time (partial observation; black), in relation to τ when using the skipping method. Dotted lines indicate individual pairs (partial observation) and runs, while solid lines indicate the mean across pairs and runs, respectively. Error bars indicate standard deviation across pairs (partial observations) and runs. (**B**) Same as (**A**), but for Φ values computed using the downsampling method. (**C**) Summary of maximal Φ values computed using the skipping method (log transformed) for each simulation (NL nonlinear; nM non-Markovian; FO full observation; PO partial observation). Triangles, circles and squares indicate log(Φ) at τ = 10, 11 and 13 ms respectively. Error bars represent standard deviation across pairs (partial observation) and runs. (**D**) Summary of maximal Φ values as in (**C**), but when computed using the downsampling method.

## Data Availability

Pre-processed fly LFPs are available on Figshare—doi: 10.26180/5ebe420ae8d89. Simulation and data analysis codes are available at https://github.com/Prototype003/phi3_timescale_sim accessed on 7 March 2022.

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
