# Peer review of "Emergence of Integrated Information at Macro Timescales in Real Neural Recordings"

_entropy, 2022, doi:10.3390/e24050625_

Round 1
Reviewer 1 Report
See attached

Reviewer 2 Report
Thank you for the opportunity to review this manuscript. I find the topic interesting, timely, and important, and while it is very complicated to get this sort of analysis just right, I think the attempt is laudable.
Overall, I find the paper well written and clear, but I do find it to rely too heavily on a prior publication in its presentation of methodology and reasoning. It is cumbersome for the reader (at least to me), and while I am sure many of my questions/concerns would be answered by studying that paper as well, it was difficult some times to know exactly which parts of the prior paper I should look to to find those answers. Personally, I believe the paper could be published more or less as is, but as a service to future readers, I would recommend adding a bit more "meat to the bone" in certain places where the authors merely cite their previous paper (see attachment).
I attach a PDF with my comments to the authors, which contains my thoughts and feedback regarding the paper. I hope the authors find some of the comments useful, but leave it to their judgement which (if any) of the comments they use to revise their manuscript. My recommendation, however, would be to at least more clearly emphasize/discuss two points: 1) The fact that they use observational data as opposed to perturbational data to construct TPMs, and 2) the fact that raw LFP amplitudes may not properly reflect the "activity" of the underlying mechanisms. These two points seem to me to be important for interpreting the results in relation to the theory.
Although I think it is possible to do quite a lot of control analyses, it is unclear to me whether they would really yield more robust findings. In the end, I have recommended "accept after minor revision", and leave it to the authors' own judgement to decide whether more extensive control analyses should be added (suggested in comments in the attached PDF).
I hope the authors find some of my suggestions useful, and if further clarifications are needed, please do not hesitate to reach out directly (I am open to discuss/clarify by email or video conference).
Congratulations, and best of luck in revising the manuscript.

Author Response
We thank the reviewer for their helpful feedback. As the methods regarding the computation of integrated information are very extensive and would add considerable bulk to the manuscript (but no new content), we decided to keep the methods as they currently are. However, we have made minor modifications to directly refer the reader first to the relevant paper, rather than to the Methods section. We hope our replies (provided in the attached PDF) and modifications have addressed the points you raised.

Round 2
Reviewer 2 Report
I have no further comments, and I am satisfied by the authors' responses to my previous comments.